# Diagnostic role of Sysmex hematology analyzer in the detection of malaria: A systematic review and meta-analysis

**Zewudu Mulatie**[1]*, **Amanuel Kelem**[2], **Elias Chane**[3], **Amare Mekuanint Tarekegn**[3], **Bisrat Birke Teketelew**[4], **Abebe Yenesew**[5], **Abateneh Melkamu**[5], **Yalew Muche**[5], **Bedasa Addisu**[2], **Dereje Mengesha Berta**[4]

1 Department of Medical Laboratory Sciences, College of Medicine and Health Sciences, Wollo University, Dessie, Ethiopia, 2 Department of Medical Laboratory Sciences, Asrat Woldeyes Health Science Campus, Debre Berhan University, Debre Berhan, Ethiopia, 3 Department of Clinical Chemistry, School of Biomedical and Laboratory Science, College of Medicine and Health Sciences, University of Gondar, Gondar, Ethiopia, 4 Department Hematology and Immunohematology, School of Biomedical and Laboratory Science, College of Medicine and Health Science, University of Gondar, Gondar, Ethiopia, 5 Department of Medical Laboratory Science, College of Medicine and Health Sciences Debre Markos University, Debre Markos, Ethiopia

* zewudumulatie@gmail.com

**Data Availability Statement:** All relevant data are within the paper.

## Abstract

### Background

Malaria control depends primarily on rapid and accurate diagnosis followed by successful treatment. Light microscopy is still used as a gold standard method for the diagnosis of malaria. The Sysmex hematology analyzer is a novel method for malaria detection. Therefore, the aim of this review was to investigate the diagnostic accuracy of the Sysmex hematology analyzer for malaria diagnosis.

### Methods

Electronic databases like PubMed, PubMed Central, Science Direct databases, Google Scholar, and Scopus were used to find relevant articles from April to June 14, 2023. The studies' methodological quality was assessed using the Quality Assessment of Diagnostic Accuracy Studies-2 tool. Using Review Manager 5.4.1, the estimates of sensitivity and specificity, as well as their 95% confidence intervals, were shown in forest plots. Midas software in Stata 14.0 was utilized to calculate the summary estimates of sensitivity, specificity, positive likelihood ratio, negative likelihood ratio, and diagnostic odds ratio. Heterogeneity was assessed by using $I^2$ statistics. In addition, publication bias was assessed using a funnel plot and Deeks' test. Sub-group and meta-regression analysis were also performed.

### Results

A total of 15 studies were assessed for diagnostic accuracy. The sensitivity and specificity of Sysmex hematology analyzer for studies ranged from 46% to 100% and 81% to 100%, respectively. The summary estimate of sensitivity and specificity of Sysmex hematology analyzer were 95% (95% CI: 85%-99%) and 99% (95% CI: 97%-100%), respectively. It had

**Funding:** The author(s) received no specific funding for this work.

**Competing interests:** The authors have declared that no competing interests exist.

**Abbreviations:** AUC, Area Under Curve; CBC, Complete Blood cell Count; DOR, Diagnostic Odd Ratio; LAMP, Loop-mediated isothermal Amplification; MI-RBC, Malaria Infected Red Blood Cell; PCR, Polymerase-Chain-Reaction; RDT, Rapid Diagnostic Test; WBC, White Blood Cell; WHO, World health organization.

excellent diagnostic accuracy. There were significant heterogeneity among the studies included in this meta-analysis. The summary estimate of sensitivity and specificity of Sysmex hematology analyzer using polymerase chain reaction as the gold standard was 97.6% (95% CI: 83.2, 99.7) and 99.4% (98.5, 99.8), respectively.

## Conclusion

In this review, Sysmex hematology analyzer had excellent diagnostic accuracy. Therefore, it could be used as an alternate diagnostic tool for malaria diagnosis in the hospital and health center.

## Trial registration

**Systematic review registration** PROSPERO (2023: CRD42023427713). https://www.crd.york.ac.uk/prospero/display_record.php?ID=CRD42023427713.

## Introduction

Malaria continues to pose a significant global public health challenge, with a significant impact on morbidity and mortality. In 2021, it was estimated that there were approximately 247 million cases of malaria infection and 619,000 reported deaths. The majority of these cases and deaths were concentrated in 29 out of the 84 countries where malaria is endemic, accounting for about 96% of global malaria cases and mortality. Particularly, Africa experienced a large proportion of the burden, with nearly 95% of cases and 96% of deaths occurring in the continent. Among these deaths, approximately 78.9% were children under the age of five. The countries who had the highest number of malaria cases were Nigeria (26.6%), the Democratic Republic of the Congo (12.3%), Uganda (5.1%), and Mozambique (4.1%), collectively contributing to almost half of all cases. Furthermore, four countries—Nigeria (31.3%), the Democratic Republic of the Congo (12.6%), the United Republic of Tanzania (4.1%), and Niger (3.9%)—accounted for slightly over half of all malaria-related deaths worldwide [1]. Early and precise diagnosis is a crucial aspect of malaria elimination efforts.

The World Health Organization (WHO) emphasizes the importance of confirming the diagnosis of suspected malaria cases through either microscopy or a rapid diagnostic test (RDT) before initiating treatment [2]. Malaria diagnosis is typically conducted by visually identifying the parasite under a microscope, which is considered the most applicable diagnostic method in clinical settings [3]. This approach is cost-effective, allows for the quantification of parasitemia, facilitates follow-up during parasitemia clearance, and offers the potential for detecting other diseases. However, microscopic diagnosis is time-consuming, requires skilled personnel, and has limitations associated with the expertise of the observer and its low limit of detection [3].

Conversely, rapid diagnostic tests (RDTs) require less training, offer convenience, and provide results within a short timeframe of approximately fifteen minutes. As a result, RDTs have effectively replaced microscopy in routine diagnosis within health institutions, as well as for active case identification by community health workers, particularly in rural areas [4]. However, RDTs do have limitations in that they are not quantitative and cannot distinguish between different stages of parasite development. Consequently, RDTs have limited utility in assessing treatment efficacy, monitoring progression, and assessing their impact on

transmission [5]. In addition, recent reports stated that the RDT cause false negatives in *Plasmodium falciparum* due to deletions of the gene encoding target histidine rich protein 2/3 (HRP2/3) [6–8]. Plasmodium lactate dehydrogenase (pLDH) detecting RDTs are also available, however they have lesser sensitivity than HRP2-detecting RDTs [9].

Nucleic acid amplification techniques (NAATs), including polymerase chain reaction (PCR), offer superior sensitivity compared to other methods, with reported detection limits as low as 5 parasites per microliter [10–12]. However, these techniques are costly and time-consuming, requiring trained personnel for operation. Additionally, they do not provide useful information for follow-up after treatment and cannot differentiate between different stages of the parasite [3]. The loop-mediated isothermal amplification (LAMP) represents a recently developed molecular technique that stands out for its simplicity, speed, and cost-effectiveness, necessitates fewer equipment and laboratory facilities compared to PCR [13]. Moreover, it exhibits higher sensitivity compared to both microscopy and RDT, making it a promising tool for efficient and accurate detection in various settings [14]. In malaria-endemic regions, there is a demand for improved diagnostic methods that can complement existing approaches, particularly those that offer high sensitivity and ease of use. The automated Sysmex hematology analyzer is an innovative and clinical tool for malaria diagnosis. This advanced technology reduces analysis time and improves accuracy in detecting potential malaria infections. It offers fast, sensitive, and cost-effective examinations for efficient malaria screening [15–17]. The analyzer works by combining optical fluorescence methods, flow cytometry, and laser-optical recognition. The Sysmex hematology analyzer provides a complete blood count (CBC) and classifies distinct blood cell types, as well as directly detects and quantifies plasmodium parasites in blood in one minute and had high a limit of quantification [16].

Several factors influence the selection of diagnostic tests, such as the cost of equipment, the number of tests to be conducted, the presence of trained personnel, the precision of the diagnostic procedure, while practicality and cost-effectiveness are crucial considerations, the paramount importance lies in the test's accuracy, even if it represents the most practical and economical option in a given scenario. Accuracy remains a critical requirement to ensure reliable diagnostic outcomes [18]. Hence, the objective of this review was to assess the diagnostic accuracy of the Sysmex hematology analyzer for malaria diagnosis.

## Methods

### Search strategy and study selection

This systematic review and meta-analysis adhered to the PRISMA guidelines for reporting. Several electronic databases, including PubMed, PubMed Central, ScienceDirect, Google Scholar, and Scopus, were searched for relevant studies from April through June 14, 2023. The search employed keywords such as "malaria," "P. falciparum," "P. vivax," "P. ovale," "P. malariae," "diagnosis," "Sysmex hematology analyzer," and "diagnostic test accuracy," with Boolean operators (AND, OR) used to combine the keywords. Overlapping studies across multiple databases were excluded. Additional papers were identified through the reference lists of the included studies. Duplicate studies and those lacking a reference test were excluded. The title and abstract of potentially acceptable research were independently screened by four reviewers. The full text of potentially eligible studies reporting the diagnostic accuracy of the Sysmex hematology analyzer was then assessed for extraction. Any disagreements among authors were resolved through discussion.

### Eligibility criteria

This review included original publications that evaluated the diagnostic accuracy of the Sysmex hematology analyzer haematology analyzer for the detection of plasmodium species. Only studies reported in English were considered. Studies reporting the diagnostic accuracy of the Sysmex hematology analyzer for the detection of malaria in non-human subjects (animals, rodents) were excluded. Review papers, case studies, and letters to the editor were also excluded.

### Data extraction and management

Data from eligible papers were independently extracted by three reviewers in Microsoft Excel 2010, including information on authors, publication year, region, study subjects, study design, descriptions of reference and index tests, and data for 2x2 tables. The methodological quality of the included studies was assessed using the QUADAS-2 tool, evaluating bias in patient selection, index test, reference standards, and flow/timing [19]. A graph illustrating the risk of bias was created.

### Statistical analysis

Statistical analyses were conducted using Review Manager 5.4 for the estimates of sensitivity and specificity, as well as their 95% confidence intervals were shown in forest plots. Midas in Stata14 used for to assess the summary estimates of sensitivity, specificity, positive likelihood ratio, negative likelihood ratio, and diagnostic odds ratio (DOR) were calculated. The hierarchical summary receiver operating characteristic curve (HSROC) was used to assess heterogeneity. Meta-disc 1.4.0 software was employed to determine the presence of a threshold effect if heterogeneity was significant or $I^2$ exceeded 50%. Meta-regression was explored to identify sources of heterogeneity, such as sampling method, study population, sample size, reference method, type of Sysmex hematology analyzer, and parameter detected. Deeks' funnel plot asymmetry test was performed to assess publication bias.

## Results

### Description of included studies

Database searches yielded a total of 936 articles, of which 612 were removed due to duplication. A total of 324 papers were screened for their titles and abstracts, and 304 studies were excluded. Among 20 full-text articles reviewed against the eligibility criteria, then 5 full-text articles were excluded. Finally, fifteen articles were eligible and included in the systematic review and meta- analysis of the final analysis (Fig 1).

### Characteristics of the included studies

The systematic review examined fifteen papers published up to 2023, resulting in 16,501 people being evaluated to evaluate the performance of the Sysmex hematology analyzer [20–33]. In 9 studies (11,220 tests), polymerase chain reaction was employed as a reference method for Sysmex hematology analyzer [20–27]. A microscope was also used as a reference method for 6 Sysmex hematology analyzer testing (5281 tests) [28–33] (Table 1).

### Data quality assessment

The risk of bias in patient selection was rated as low in six of the fifteen diagnostic studies and high in two of the fifteen studies. In the applicability concern domain, there was no high risk

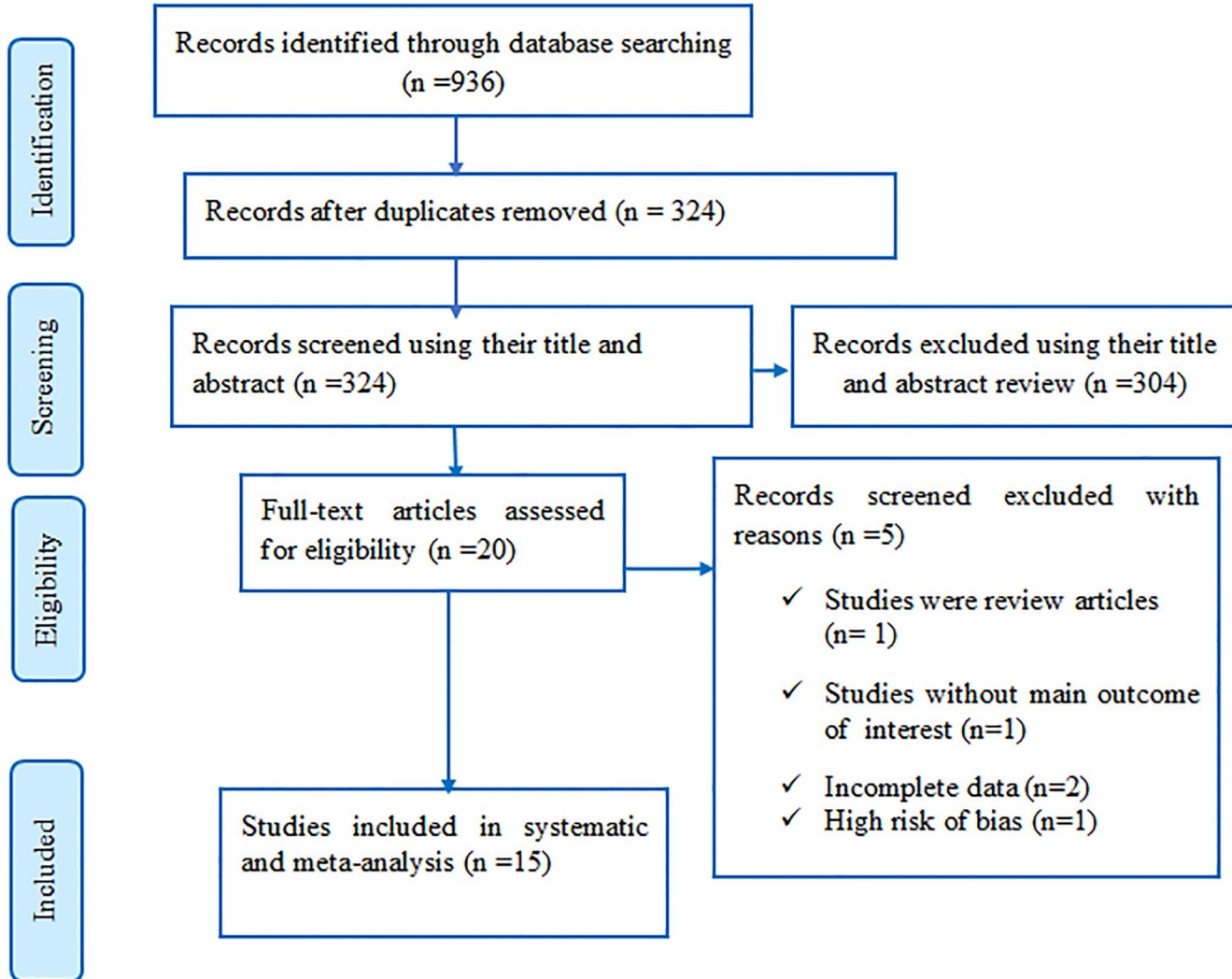

**Fig 1. Flow diagram of the included studies for the systematic review and meta-analysis.**

of bias in patient selection. The risk of bias in flow and timing was assessed as low in the nine studies (Fig 2).

### Diagnostic accuracy of Sysmex hematology analyzer

The systematic review and meta-analysis comprised fifteen studies to evaluate the diagnostic accuracy of Sysmex hematology analyzer. For those studies, the sensitivity and specificity ranged from 46% to 100% and 81% to 100%, respectively. For those investigations, the summary estimates of sensitivity and specificity of Sysmex hematology analyzer were 95% (95%, CI: 85% -99%) and 99% (95%, CI: 97% -100%), respectively. Sysmex hematology analyzer had a DOR of 1890 (95% CI: 392–9112), a positive likelihood ratio (LR+) of 94.1 (95% CI: 34.6–256.1), and a negative likelihood ratio (LR-) of 0.05 (95% CI: 0.03–0.07). The area under curve (AUC) was 1.00 (95% CI: 0.99–1.00), indicating that the test had excellent diagnostic accuracy (Fig 3).

**Table 1. Characteristics of the studies included in the systematic review and meta-analysis of the performance of Sysmex hematology analyzer.**

| s. no. | Author (year) | Population | Age in years ((median (IQR)) | parasite density ((median (IQR)) | Sample size | TP | FP | TN | FN | Sysmex type | Reference test | Reference |
|---|---|---|---|---|---|---|---|---|---|---|---|---|
| 1 | M'baya et al (2022) [20] | Malaria suspected | NA | 164 (63–448) | 5031 | 513 | 41 | 4477 | 0 | XN-31 | PCR | [20] |
| 2 | Buoro et al (2018) [28] | Malaria suspected | 12–64* | NA | 1061 | 13 | 200 | 847 | 1 | XN series | microscope | [28] |
| 3 | Huh et al (2008) [29] | Patients (asymptomatic) | NA | 4003.3±6530.2** | 487 | 100 | 0 | 343 | 44 | XE-2100 | microscope | [29] |
| 4 | Kagaya et al (2022) [21] | Malaria suspected | 23 (5–36) | 21,774 (9,490 –101,973) | 169 | 18 | 11 | 135 | 5 | XN-31 | PCR | [21] |
| 5 | Khartabil et al (2022) [30] | Malaria suspected | NA | NA | 112 | 13 | 1 | 96 | 2 | XN-31 | microscope | [30] |
| 6 | Picot et al (2022) [22] | Febrile patients | 36 (1–89) | 2.4–288.3* | 357 | 109 | 4 | 244 | 0 | XN-31 | PCR | [22] |
| 7 | Pillay et al 7[a] (2019) [23] | Malaria suspected | NA | NA | 191 | 124 | 0 | 67 | 0 | XN-30 | PCR | [23] |
| 8 | Pillay et al 7[b](2019) [23] | Malaria suspected | NA | NA | 1028 | 272 | 1 | 749 | 6 | XN-10 | PCR | [23] |
| 9 | Post et al (2019) [24] | Febrile patients | 0.67–59* | 12,390 (650–88,656) | 837 | 253 | 2 | 478 | 104 | XN-30 | PCR | [24] |
| 10 | Mohapatra et al (2011) [31] | Malaria suspected | NA | NA | 430 | 52 | 35 | 325 | 18 | XE-2100 | microscope | [31] |
| 11 | Sharma et al (2013) [32] | Malaria suspected | 5–60* | NA | 2251 | 147 | 129 | 1974 | 1 | XE-2000i | microscope | [32] |
| 12 | Sunilkumar et al (2016) [33] | Febrile patients | 1–65* | NA | 940 | 40 | 24 | 867 | 9 | XN 1000 | microscope | [33] |
| 13 | Yasuda et al (2022) [25] | Malaria suspected | NA | NA | 80 | 32 | 0 | 47 | 1 | XN-31 | PCR | [25] |
| 14 | Yoo et al (2010) [26] | Patients (asymptomatic) | NA | 10,682.3 ± 3,458.2** | 1801 | 191 | 4 | 1384 | 222 | XE-2100 | PCR | [26] |
| 15 | Zuluaga et al (2021) [27] | Febrile patients | 27 (14–44) | 3508 (886–10613) | 1726 | 513 | 2 | 1154 | 57 | XN-31 | PCR | [27] |

Note: NA, not applicable; PCR: Polymerase chain reaction; TP, true positive; FP, false positive; FN, false negative; TN: True negative; MI-RBC: Malaria infected red blood cell; Diff-WBC: Differential white blood cell; IQR: Inter quartile range.

NB; [a] or [b]: Indicates in one study compare 2 types of symex hematology analyzer; NA: Indicates the result not available;

*: Indicates the result reported by range;

**: Indicates the result reported by mean and standard deviation.

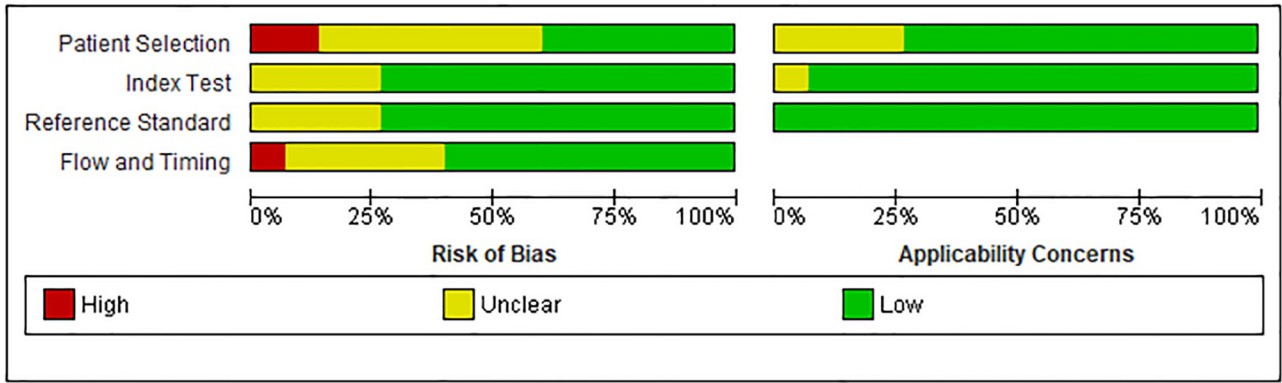

**Fig 2. Risk of bias graph of studies included in the meta-analysis.**

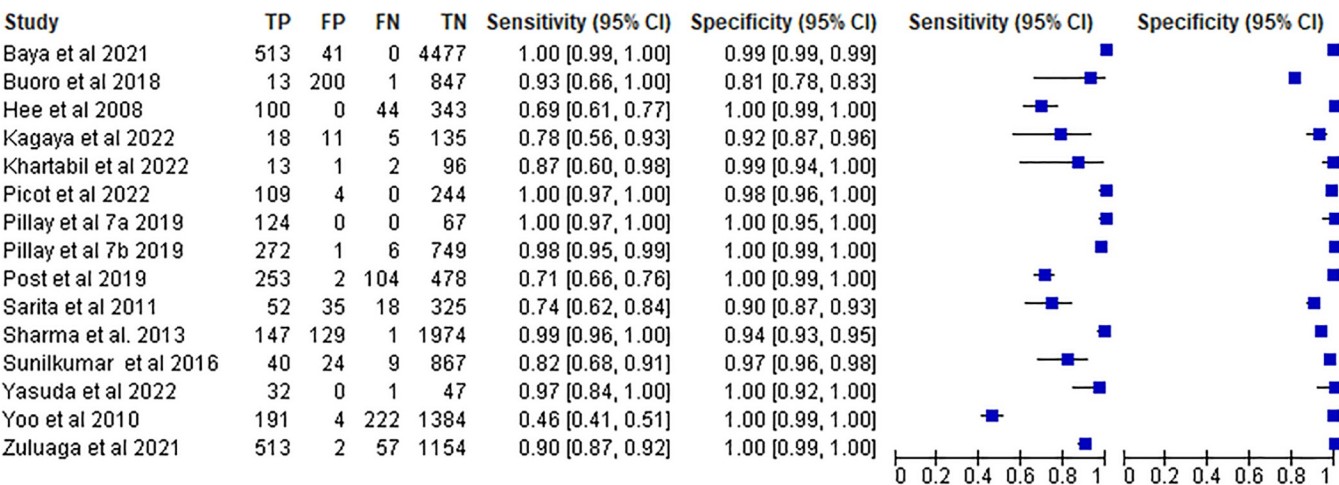

| Study | TP | FP | FN | TN | Sensitivity (95% CI) | Specificity (95% CI) |
|---|---|---|---|---|---|---|
| Baya et al 2021 | 513 | 41 | 0 | 4477 | 1.00 [0.99, 1.00] | 0.99 [0.99, 0.99] |
| Buoro et al 2018 | 13 | 200 | 1 | 847 | 0.93 [0.66, 1.00] | 0.81 [0.78, 0.83] |
| Hee et al 2008 | 100 | 0 | 44 | 343 | 0.69 [0.61, 0.77] | 1.00 [0.99, 1.00] |
| Kagaya et al 2022 | 18 | 11 | 5 | 135 | 0.78 [0.56, 0.93] | 0.92 [0.87, 0.96] |
| Khartabil et al 2022 | 13 | 1 | 2 | 96 | 0.87 [0.60, 0.98] | 0.99 [0.94, 1.00] |
| Picot et al 2022 | 109 | 4 | 0 | 244 | 1.00 [0.97, 1.00] | 0.98 [0.96, 1.00] |
| Pillay et al 7a 2019 | 124 | 0 | 0 | 67 | 1.00 [0.97, 1.00] | 1.00 [0.95, 1.00] |
| Pillay et al 7b 2019 | 272 | 1 | 6 | 749 | 0.98 [0.95, 0.99] | 1.00 [0.99, 1.00] |
| Post et al 2019 | 253 | 2 | 104 | 478 | 0.71 [0.66, 0.76] | 1.00 [0.99, 1.00] |
| Sarita et al 2011 | 52 | 35 | 18 | 325 | 0.74 [0.62, 0.84] | 0.90 [0.87, 0.93] |
| Sharma et al. 2013 | 147 | 129 | 1 | 1974 | 0.99 [0.96, 1.00] | 0.94 [0.93, 0.95] |
| Sunilkumar et al 2016 | 40 | 24 | 9 | 867 | 0.82 [0.68, 0.91] | 0.97 [0.96, 0.98] |
| Yasuda et al 2022 | 32 | 0 | 1 | 47 | 0.97 [0.84, 1.00] | 1.00 [0.92, 1.00] |
| Yoo et al 2010 | 191 | 4 | 222 | 1384 | 0.46 [0.41, 0.51] | 1.00 [0.99, 1.00] |
| Zuluaga et al 2021 | 513 | 2 | 57 | 1154 | 0.90 [0.87, 0.92] | 1.00 [0.99, 1.00] |

**Fig 3. Forest plot of Sysmex hematology analyzer for malaria detection.**

## Sub group analysis

Studies conducted to assess Sysmex hematology analyzer accuracy revealed significant heterogeneity (Q = 416.938, I2 = 100, and P< 0.01). Sub-group analysis was performed based on continent, publication year, study design, study population, reference tests, Sysmex hematology analyzer types, and malaria detection parameters. Pooled sensitivity in febrile patients was 92% (95% CI: 65%, 99%), while pooled sensitivity in malaria suspected patients was 97% (95% CI: 82%, 99%). When PCR was employed as a reference test, Sysmex hematology analyzer had sensitivity and specificity of 98% (95% CI: 83, 100) and 99% (99, 100), respectively. When the microscope test was used as the gold standard, the sensitivity and specificity of Sysmex hematology analyzer were 82% (95% CI: 44%, 96%) and 97% (95% CI: 89%, 99%), respectively. The sensitivity of the Sysmex hematology analyzer XN series was 98% (90%, 99%), while the sensitivity of the Sysmex hematology analyzer XE series was 83% (95%CI: 44%, 97%) using of PCR and microscope as gold standard. The threshold effect and meta-regression were used to investigate the cause of heterogeneity. There was no threshold effect between Asian studies (P = 0.19), between 2016 and 2020 (P = 0.28), febrile patients (P = 0.07), Sysmex hematology analyzer with microscope (P = 0.32), Sysmex hematology analyzer XE series (P = 0.33), mi-RBC (P = 0.21), and diff-WBC (P = 0.38) (Table 2).

The meta-regression of those studies revealed that the most significant sources of heterogeneity (P<0.05) were the reference test, Sysmex hematology analyzer types, and malaria detection parameters (Table 3).

## Sysmex hematology analyzer with PCR and microscope as reference test

The sensitivity and specificity of Sysmex hematology analyzer performed with PCR as a reference test were varies from 46%-100% and 92%-100%, respectively. Six of the studies (66.7%, 6/9) demonstrated a sensitivity of 90% and more. Eight (88.9%, 8/9) studies showed a specificity ≥95%. The sensitivity and specificity of Sysmex hematology analyzer done with microscope as reference test ranges from 69%–99% to 81%–100%, respectively. Two (2/6) studies showed a sensitivity ≥90%. Three (50%, 3/6) studies showed a specificity ≥95% (Fig 4).

The summary estimate of sensitivity and specificity of Sysmex hematology analyzer were 97.6% (95% CI: 83.2, 99.7) and 99.4% (95% CI: 98.5, 99.8), respectively, using PCR as reference

**Table 2. Subgroup analysis of Sysmex hematology analyzer.**

| Sub group | | No. of studies | Pooled sensitivity (95%CI) | Pooled specificity (95%CI) | Heterogeneity test ($I^2$) | P-value |
|---|---|---|---|---|---|---|
| Continent | Africa | 5 | 99%(75%, 100%) | 99% (0.97%, 100%) | 97% | < 0.01* |
| | Asia | 6 | 86% (62%, 96%) | 99% (94%, 100% | 99% | 0.19 |
| | Total pooled | 11 | 94% (79%,98) | 99 (97%, 100%) | 99% | <0.01* |
| Publication year | ≤ 2015 | 4 | 83% (44%, 97%) | 99% (91%, 100%) | 99% | 0.33 |
| | 2016–2020 | 5 | 95% (75%, 99%) | 99% (92%, 100%) | 95% | 0.28 |
| | 2021–2023 | 6 | 97% (55%, 100%) | 99% (97%, 100%) | 98% | < 0.01* |
| | Total pooled | 15 | 95% (85%, 99% | 99% (97%, 100%) | 100% | < 0.01* |
| Study design | Cross sectional | 10 | 87% (63%, 96%) | 99% (97%, 100%) | 98% | 0.04* |
| | Prospective | 4 | 99% (45%, 100%) | 98% (95%, 100%) | 98% | < 0.01* |
| | Total pooled | 14 | 94% (77%, 99%) | 99% (98, 100%) | 99% | < 0.01* |
| Population | Febrile patient | 4 | 92% (65%, 99%) | 99% (98%, 100%) | 91% | 0.07 |
| | Malaria suspected | 9 | 97% (82%, 99%) | 98% (95%, 99%) | 93% | 0.04* |
| | Total pooled | 13 | 96%(85%, 99) | 98% (97%, 99%) | 95% | 0.03* |
| Reference tests | PCR | 9 | 98% (83, 100) | 99 (99, 100) | 99% | < 0.01* |
| | Microscope | 6 | 82% (44%, 96%) | 97% (89%, 99% | 99% | 0.32 |
| | Total pooled | 15 | 95% (85%, 99% | 99% (97%, 100%) | 100% | < 0.01* |
| Sysmex hematology analyzer types | XN series | 11 | 98% (90%, 99%) | 99% (98%, 100%) | 98% | 0.02* |
| | XE series | 4 | 83% (44%, 97%) | 99% (91%, 100) | 99% | 0.33 |
| | Total pooled | 15 | 95% (85%, 99% | 99% (97%, 100%) | 100% | < 0.01* |
| Parameters | mi-RBC | 10 | 96% (77%, 99%) | 99% (98%, 99%) | 97% | 0.21 |
| | Diff WBC | 5 | 86% (56%, 97%) | 98% (86%, 100%) | 99% | 0.38 |
| | Total pooled | 15 | 95% (85%, 99% | 99% (97%, 100%) | 100% | < 0.01* |

Note;

*statistically significant.

test. It had a DOR of 7880.01 (95% CI: 791.03, 78498.74), a positive likelihood ratio (LR+) of 186.87 (95% CI: 67.93, 514.07), and a negative likelihood ratio (LR-) of 0.023 (95% CI: 0.003, 0.188). The area under the curve was 1.00 (95% CI: 0.99–1.00), indicating that the test had excellent diagnostic accuracy (Fig 5).

## Sysmex hematology analyzer with microscope as reference

The summary estimate of sensitivity and specificity of employing a microscope as the gold standard method were estimated to be 81.7% (95% CI: 44%, 96) and 97.02% (95% CI: 88.92,

**Table 3. Meta-regression analysis of diagnostic accuracy.**

| Variables | Coefficient | Standard error | p-value | RDOR | 95%CI |
|---|---|---|---|---|---|
| Ste | 6.12 | 1.41 | <0.01 | NA | NA |
| S | 0.43 | 0.12 | 0.03 | NA | NA |
| Study population | -0.08 | 0.76 | 0.89 | 0.65 | 0.12–5.31 |
| Sample size | −1.45 | 0.57 | 0.03* | 0.49 | 0.02–0.94 |
| Reference test | 1.09 | 0.3541 | 0.04* | 2.15 | 1.19–19.03 |
| Sysmex hematology analyzer types | 1.67 | 0.5436 | 0.02* | 3.46 | 1.32–15.91 |
| Parameters | 1.81 | 0.6310 | 0.01* | 4.29 | 1.61–19.67 |

*statistically significant NA, not applicable; CI, confidence interval; RDOR, relative diagnostic odds ratio; Ste, constant term in the equation; S, a measure of threshold.

sysmex with PCR

| Study | TP | FP | FN | TN | Sensitivity (95% CI) | Specificity (95% CI) |
|-------|-----|-----|-----|------|----------------------|----------------------|
| Baya et al 2021 | 513 | 41 | 0 | 4477 | 1.00 [0.99, 1.00] | 0.99 [0.99, 0.99] |
| Kagaya et al 2022 | 18 | 11 | 5 | 135 | 0.78 [0.56, 0.93] | 0.92 [0.87, 0.96] |
| Picot et al 2022 | 109 | 4 | 0 | 244 | 1.00 [0.97, 1.00] | 0.98 [0.96, 1.00] |
| Pillay et al 7a 2019 | 124 | 0 | 0 | 67 | 1.00 [0.97, 1.00] | 1.00 [0.95, 1.00] |
| Pillay et al 7b 2019 | 272 | 1 | 6 | 749 | 0.98 [0.95, 0.99] | 1.00 [0.99, 1.00] |
| Post et al 2019 | 253 | 2 | 104 | 478 | 0.71 [0.66, 0.76] | 1.00 [0.99, 1.00] |
| Yasuda et al 2022 | 32 | 0 | 1 | 47 | 0.97 [0.84, 1.00] | 1.00 [0.92, 1.00] |
| Yoo et al 2010 | 191 | 4 | 222 | 1384 | 0.46 [0.41, 0.51] | 1.00 [0.99, 1.00] |
| Zuluaga et al 2021 | 513 | 2 | 57 | 1154 | 0.90 [0.87, 0.92] | 1.00 [0.99, 1.00] |

sysmex with microscope

| Study | TP | FP | FN | TN | Sensitivity (95% CI) | Specificity (95% CI) |
|-------|-----|-----|-----|------|----------------------|----------------------|
| Buoro et al 2018 | 13 | 200 | 1 | 847 | 0.93 [0.66, 1.00] | 0.81 [0.78, 0.83] |
| Hee et al 2008 | 100 | 0 | 44 | 343 | 0.69 [0.61, 0.77] | 1.00 [0.99, 1.00] |
| Khartabil et al 2022 | 13 | 1 | 2 | 96 | 0.87 [0.60, 0.98] | 0.99 [0.94, 1.00] |
| Sarita et al 2011 | 52 | 35 | 18 | 325 | 0.74 [0.62, 0.84] | 0.90 [0.87, 0.93] |
| Sharma et al. 2013 | 147 | 129 | 1 | 1974 | 0.99 [0.96, 1.00] | 0.94 [0.93, 0.95] |
| Sunilkumar et al 2016 | 40 | 24 | 9 | 867 | 0.82 [0.68, 0.91] | 0.97 [0.96, 0.98] |

**Fig 4. Forest plot of Sysmex hematology analyzer with PCR and microscope as reference test.**

99.7), respectively. With a microscope as the gold standard, the diagnostic odds ratio, LR+, and LR- of Sysmex were 146.27 (95% CI: 29.86, 716.4), 27.50 (95% CI: 8.03, 94.20), and 0.188 (95% CI: 0.046, 0.763), respectively. The AUC was 0.98 (95% CI: 0.96–0.99), indicating that the test had excellent diagnostic accuracy (Fig 6).

## Publication bias

The Deeks' funnel plot asymmetry test of DOR revealed no significant asymmetry (P = 0.14), indicating that there was no observable publication bias (Fig 7).

## Discussion

An accurate and timely diagnosis is critical for managing, controlling, and eradicating malaria. Early malaria diagnosis is necessary to guide adequate treatment and decrease the severe effects of infection [1]. Lack of clinical and laboratory experience, prolonged incubation periods and *Plasmodium vivax* relapses [34,35], or prophylaxis in travellers can all delay diagnosis, increasing the risk of malaria-related complications [36]. An accurate malaria detection technology integrated into the routine Sysmex hematology analyzer could assist detect infections earlier and potentially reduce complications of malaria associated with malaria infection. It also utilized by minimally trained personnel, does not require sample preparation, has no observer-dependent variability, and is available in the hospital and health center. Another significant advantage of the Sysmex hematology analyzer automation is that it delivers a CBC with each assay, integrating malaria diagnosis and treatment [15].

All accessible studies, both symptomatic and asymptomatic, were extensively searched. The majority of these investigations were undertaken after 2015; only one study was conducted before 2010, reflecting the relatively recent interest in this diagnostic approach. The $I^2$ statistic was used for investigating statistical heterogeneity, which reflects variance across studies due

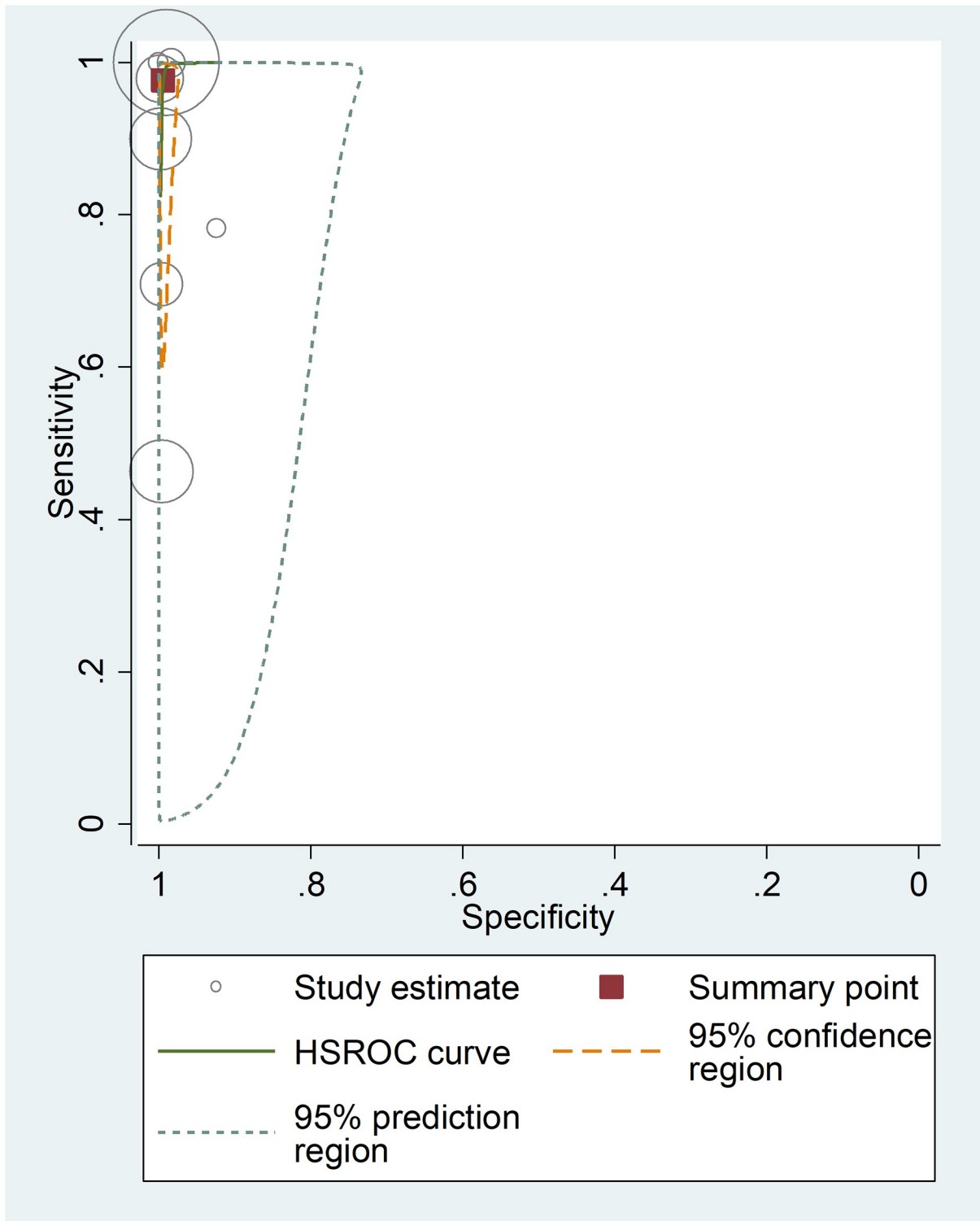

**Fig 5. Summary receiver operating characteristic plot of sensitivity and specificity of Sysmex hematology analyzer with PCR as a reference test.**

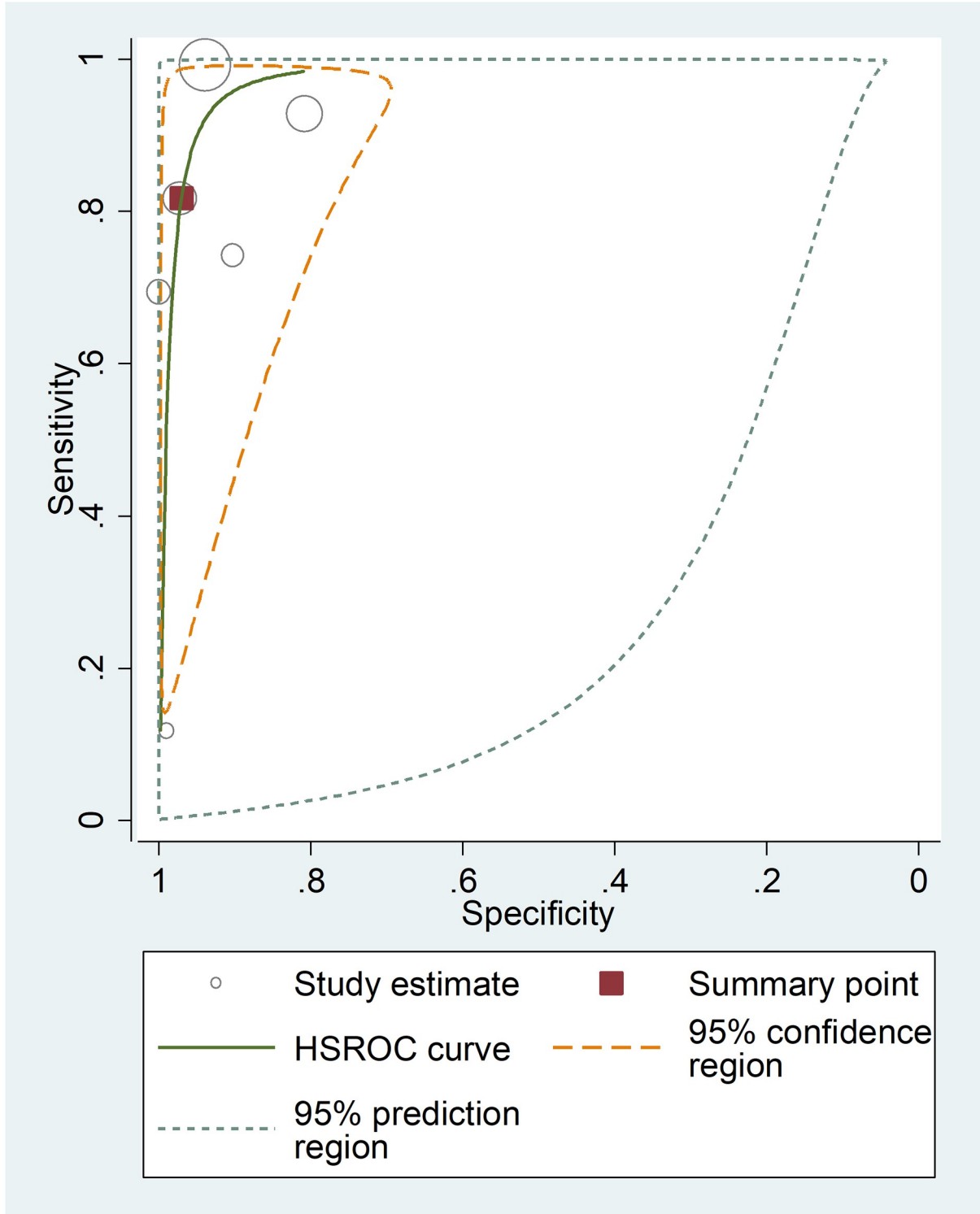

**Fig 6. Summary receiver operating characteristic plot of sensitivity and specificity of Sysmex hematology analyzer with microscope as a reference test.**

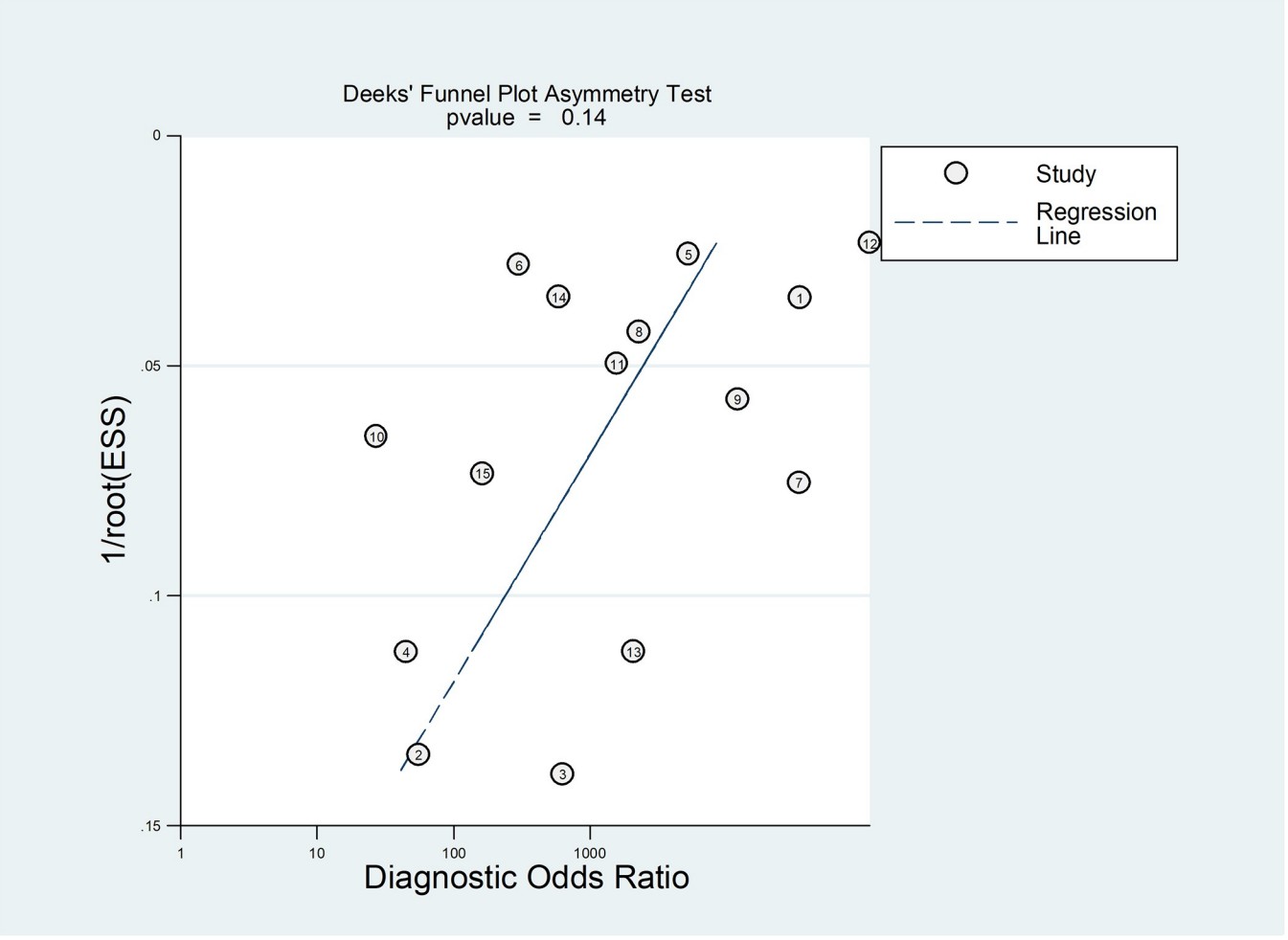

**Fig 7. Publication bias of Sysmex hematology analyzer for diagnostic accuracy.**

to inter-study variability. Heterogeneity was expected to be associated with the test reading method, the level of parasitaemia, and the comparators. There was significant heterogeneity among the studies included in this meta-analysis. Meta-regression analysis for the Sysmex hematology analyzer revealed that the gold standard reference test, the Sysmex hematology analyzer types and parameters for malaria detection were the sources of heterogeneity.

The overall estimated sensitivity and specificity of Sysmex hematology analyzer were 95% (95%CI: 85–99%) and 99% (95%CI: 97%-100%), respectively, according to this review. The Sysmex hematology analyzer had high diagnostic accuracy (AUC: 1.00 (95% CI: 0.99–1.00). Sysmex hematology analyzer had 1890 (95% CI 392–9112) times higher odds of obtaining a positive result in a diseased person than in a non-diseased one. On the other hand, it showed 94.1 (95% CI: 34.6–256.1) times higher odds of a positive individual and a negative one. Therefore, the Sysmex hematology analyzer is employed as an alternative diagnostic tool for malaria, given its high diagnostic accuracy.

This review's pooled sensitivity was higher than the sensitivity of studies on RDT diagnostic accuracy (42% for conventional vs. 61% for ultrasensitive RDT) [37], while another study revealed lower sensitivity (44.3% for conventional and 56.1% for high sensitive RDT) [38]. The

reduced sensitivity of RDT may be attributed to asymptomatic malaria and deletions of the *Plasmodium falciparum* gene encoding the target histidine rich protein 2/3 (HRP2/3). The microscopic diagnosis of malaria tends to be less sensitive in low parasitaemia and has been reported as 70% in pregnancy-associated malaria [39]. It could be because microscopes require expert laboratory personnel, have limited diagnostic accuracy, and generate false negative results in asymptomatic malaria. This review's pooled sensitivity was consistent with a systematic review and meta-analysis of the diagnostic accuracy of loop-mediated isothermal amplification (LAMP) techniques, which indicated a sensitivity of 97% [40] and another review, which reported the sensitivity of PCR to be 98% [41]. The pooled specificity of Sysmex hematology analyzer had high specificity in this review and was almost identical to the specificity of RDT [37], microscope [39], LAMP [40] and PCR [41].

In this review, Sysmex hematology analyzer with PCR as the gold standard was shown to have a higher pooled sensitivity than Sysmex hematology analyzer with microscopy as gold standard (97.6% (95% CI: 83.2–99.7%) vs. 81.7% (95% CI: 44%, 96). However, the overall estimate of specificity of Sysmex hematology analyzer using PCR as the reference method and Sysmex hematology analyzer utilising microscopy as the gold standard method had comparable specificity (99.4% (95% CI: 98.5, 99.8) vs. 97.02% (95% CI: 88.92, 99.7%), respectively. Also, the diagnostic accuracy of Sysmex hematology analyzer with PCR as gold standard and Sysmex hematology analyzer with microscope was almost similar (AUC: 1.00 (95% CI: 0.99–1.00) vs. 0.98 (95% CI: 0.96–0.99)), respectively.

The sensitivity of Sysmex hematology analyzer with PCR as gold standard was greater in this review than a systematic review and meta-analysis study on microscope with PCR in Ethiopia, which showed a sensitivity of 75.20% [18], and another review in Colombia, which stated a sensitivity of 70.8% [39]. In the other hand, the specificity of this study was nearly identical to that of a studies on microscopy with PCR in Ethiopia (97.12%) [18] and Colombia (99%) [39]. The summary estimates of sensitivity this review higher than the review done about RDT with PCR in Ethiopia were 66.18% [18]. In contrast, the summary estimations of specificity of this review were similar to review done in Ethiopia (95.36%) [18]. This review's sensitivity was almost identical to that of a study on LAMP using PCR as gold standard (97%) [40].

The sensitivity of Sysmex hematology analyzer with microscope as reference test was lower in this review than in a systematic review and meta-analysis study on RDT with microscope in Ethiopia (95.05%) [18] and India (97.0%) [42]. The lower sensitivity for this review could be attributed to the fact that the majority of the studies used differential WBC scatter for malaria diagnosis. According to a review of haematology analyzers for malaria detection, the decrease in sensitivity for differential WBC abnormalities could be due to the lack of a consensus definition for this diagnosis criterion, as well as difficulty and subjectivity in manually evaluating these patterns, which could have resulted in a classification bias. However, the specificity of this study was nearly same to that of review conducted in Ethiopia (96.47%) [18] and India (96.0%) [42].

The sensitivity of Sysmex hematology analyzer XN series was higher compared to XE series (98% vs. 83%), respectively in this review. The Sysmex hematology analyzer XN series and XE series types had similar specificity (99%). Both Sysmex hematology analyzer XN series and XE series types had excellent diagnostic accuracy (AUC: 0.99 (95%CI: 0.99–1.00) vs. 0.99 (95%CI: 0.97–0.99)).

In this review, the sensitivity of Sysmex hematology analyzer by using malaria infected RBC for malaria detection was higher than the use of diff WBC (96% vs. 86%), respectively. The Sysmex hematology analyzer using mi-RBC and diff-WBC for malaria detection had almost similar specificity (99% vs.98%), respectively. Generally, both Sysmex hematology analyzer with

mi- RBC and diff- WBC techniques had excellent diagnostic accuracy (AUC: 0.99 (95%CI: 0.98–1.00) vs. 0.98 (95%CI: 0.96–0.99)).

## Conclusion

Sysmex hematology analyzer had high summary estimate of sensitivity and specificity. The pooled sensitivity was higher for Sysmex hematology analyzer using PCR as the gold standard than for Sysmex hematology analyzer using a microscope as the gold standard. As a result, Sysmex hematology analyzer is used as an alternate diagnostic tool for malaria diagnosis in malaria endemic area and non-endemic area due to its high diagnostic accuracy. The Sysmex hematology analyzer boasts numerous implications. It enabling the detection of parasites in both endemic and non-endemic areas. Furthermore, it is capable of simultaneously providing CBC results, identifying parasites even at low concentrations, and identifying different stages of the parasite. the ability to quantify parasite density and monitor treatment response. The analyzer requires minimal training, does not require sample preparation, has no observer-dependent variability and comes at a cost comparable to that of performing a CBC. Additionally, it can be utilized in blood banks to screen for malaria. However, its application in rural settings or post-health care scenarios presents challenges, as it necessitates stable electricity and a reliable supply of reagents, making it difficult to replace RDTs in these specific contexts.

## Supporting information

**S1 File. PRISMA checklist for "Diagnostic role of Sysmex hematology analyzer in the detection of malaria: A systematic review and meta-analysis.**
(DOCX)

## Acknowledgments

The authors would like to express their gratitude to the authors of the included research as well as the participants in their investigations.

## Author Contributions

**Conceptualization:** Zewudu Mulatie, Amanuel Kelem, Elias Chane, Dereje Mengesha Berta.

**Data curation:** Zewudu Mulatie, Amare Mekuanint Tarekegn, Bisrat Birke Teketelew, Yalew Muche.

**Formal analysis:** Zewudu Mulatie, Amanuel Kelem, Elias Chane, Bisrat Birke Teketelew, Abebe Yenesew, Abateneh Melkamu, Yalew Muche, Bedasa Addisu, Dereje Mengesha Berta.

**Funding acquisition:** Zewudu Mulatie, Bedasa Addisu.

**Investigation:** Zewudu Mulatie, Amare Mekuanint Tarekegn, Abebe Yenesew, Abateneh Melkamu.

**Methodology:** Zewudu Mulatie, Amanuel Kelem, Bisrat Birke Teketelew, Abateneh Melkamu, Yalew Muche.

**Project administration:** Zewudu Mulatie, Amanuel Kelem, Dereje Mengesha Berta.

**Resources:** Zewudu Mulatie, Elias Chane.

**Software:** Zewudu Mulatie, Amanuel Kelem, Amare Mekuanint Tarekegn, Bisrat Birke Teketelew, Abebe Yenesew, Abateneh Melkamu, Yalew Muche, Bedasa Addisu, Dereje Mengesha Berta.

**Supervision:** Zewudu Mulatie, Amanuel Kelem, Abebe Yenesew, Dereje Mengesha Berta.

**Validation:** Zewudu Mulatie, Amare Mekuanint Tarekegn, Abateneh Melkamu, Yalew Muche.

**Visualization:** Zewudu Mulatie, Elias Chane, Abateneh Melkamu, Bedasa Addisu.

**Writing – original draft:** Zewudu Mulatie, Amanuel Kelem, Elias Chane, Amare Mekuanint Tarekegn, Bisrat Birke Teketelew, Abebe Yenesew, Bedasa Addisu, Dereje Mengesha Berta.

**Writing – review & editing:** Zewudu Mulatie, Amanuel Kelem, Elias Chane, Amare Mekuanint Tarekegn, Bisrat Birke Teketelew, Abebe Yenesew, Abateneh Melkamu, Yalew Muche, Bedasa Addisu, Dereje Mengesha Berta.

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
