## [Decision Letter · Decision Letter 0]

15 Nov 2023

PONE-D-23-30250Diagnostic role of Sysmex hematology analyzer in the detection of malaria: A systematic review and meta-analysisPLOS ONE

Dear Dr. Mulatie,

Thank you for submitting your manuscript to PLOS ONE. After careful consideration, we feel that it has merit but does not fully meet PLOS ONE’s publication criteria as it currently stands. Therefore, we invite you to submit a revised version of the manuscript that addresses the points raised during the review process.

We look forward to receiving your revised manuscript.

Kind regards,

Bosco Bekiita Agaba, PhD

Academic Editor

PLOS ONE

Journal Requirements:

https://malariajournal.biomedcentral.com/articles/10.1186/s12936-021-03923-8

https://www.nature.com/articles/s41598-021-84594-y?code=9237a5ab-4dae-47c8-b21b-0d7c9c1b2254&error=cookies_not_supported

3. In your revision ensure you cite all your sources (including your own works), and quote or rephrase any duplicated text outside the methods section. Further consideration is dependent on these concerns being addressed.

4. We suggest you thoroughly copyedit your manuscript for language usage, spelling, and grammar. If you do not know anyone who can help you do this, you may wish to consider employing a professional scientific editing service. 

Whilst you may use any professional scientific editing service of your choice, PLOS has partnered with both American Journal Experts (AJE) and Editage to provide discounted services to PLOS authors. Both organizations have experience helping authors meet PLOS guidelines and can provide language editing, translation, manuscript formatting, and figure formatting to ensure your manuscript meets our submission guidelines. To take advantage of our partnership with AJE, visit the AJE website (http://aje.com/go/plos) for a 15% discount off AJE services. To take advantage of our partnership with Editage, visit the Editage website (www.editage.com) and enter referral code PLOSEDIT for a 15% discount off Editage services. If the PLOS editorial team finds any language issues in text that either AJE or Editage has edited, the service provider will re-edit the text for free.

7. Please amend either the abstract on the online submission form (via Edit Submission) or the abstract in the manuscript so that they are identical.

Additional Editor Comments:

General:

Can one load a blood sample to test for malaria only? Or one has to run a CBC to look for parasites alongside a CBC?

What happens when a patient only wants malaria test alone, or if this is not possible, What’s the economic implication if one has to run a malaria test alongside a CBC for malaria patients?

At what level of care within the system would the equipment be suitable for optimal utilization?

Despite the challenges faced by RDTs, they are easy to use and can be used by all cadres as POCs including nurses, Lab techs, etc hence increasing access to parasite-based diagnosis; based on this, What would be health cadre to use this technology for malaria diagnosis in terms qualification (scientists, Lab techs, nurses or other cadres trained to test for malaria) and how does this affect access to testing and affect the equipment advantage?

Were these mostly done in Africa where the malaria burden is, what was the distribution of the reviewed studies?

How many types or versions of sysmex equipment are able to do malaria testing? Did the studies cover all versions or one equipment version and what’s the implication?

What patient populations (symptomatic, asymptomatic, age, parasite densities, etc, could u provide a table summarizing patient baseline characteristics?

Line: 108: revise the sentience- uantity?

Line 132-33: How many non-English articles/studies were excluded? Unpublished reports and conference papers?

Line 159: whats the authors comment on using a sample of 15 papers for a systematic review to draw the conclusions?

Line 165-67: what was the reference/gold standard?

Reviewers' comments:

Reviewer's Responses to Questions

**Comments to the Author**

1. Is the manuscript technically sound, and do the data support the conclusions?

Reviewer #1: Partly

Reviewer #2: Yes

Reviewer #3: Yes

2. Has the statistical analysis been performed appropriately and rigorously? 

Reviewer #1: Yes

Reviewer #2: Yes

Reviewer #3: Yes

3. Have the authors made all data underlying the findings in their manuscript fully available?

Reviewer #1: Yes

Reviewer #2: Yes

Reviewer #3: Yes

4. Is the manuscript presented in an intelligible fashion and written in standard English?

Reviewer #1: Yes

Reviewer #2: Yes

Reviewer #3: Yes

5. Review Comments to the Author

Reviewer #1: General comments:

-The study's objective is to assess Sysmex's diagnostic performance in diagnosing malaria; however, the cost was not considered by the authors. Cost is quite important, particularly in areas where malaria is a major problem.

-Where this device will be used was not specified by the authors. Is it in the comprehensive hospital, primary hospital, or health center? Can this device be used in health post-level settings to replace the existing RDT?

-The authors advised to revise the conclusion section. The conclusion fails to persuade the readers based on the findings.

-Apart from PCR, the diagnostic performance of LAMP is extensively studied and recommended by most scientists in the fields. Nevertheless, the authors didn't mentioned about LAMP in the background section.

-Control and eradication were terms employed by the authors in the body of the manuscript. They overlooked a critical step "elimination". The elimination process is crucial to the activities of a particular country or subregional program. For instance, in Ethiopia (where the current program is malaria elimination) or other East African countries where gene deletions are currently highly prevalent, this method may be suggested for the diagnosis of malaria.

-The authors advised italicizing every malaria species mentioned in the text's main body.

-In the background the cost implication of using PCR was not mentioned.

-What more advantages come with using a Sysmex hematological analyzer than those already mentioned?

-The year of studies indicated in the abstract and the paper body are different, so please correct them.

-Table-1 need revisions. It would be nice to include one column for reference.

Specific comments:

-Line 176, in the nine studies are ?? needs revision.

-Line 276. "Because of has high diagnostic accuracy" needs revision.

-Table 2 needs formatting.

-The conclusion must be changed in line with the findings.

Reviewer #2: Please revise grammar and spelling errors, I am listing some here. I have also suggested some options to paraphrase some sentences for clarity.

• 108: uantity =quantity

• 214: significant, confidence

• 237: Sysmex

• 253: malaria-related complications

• 254: into the routine

• 252: plasmodium=Plasmodium

• 255: instead of “negative consequences…” consider “complications of malaria”.

• 256: not requires = does not require

• 264: associated with

• 265: consider replacing “ the patient, parasitaemia “ with “ the level of parasitaemia”.

• 266-267: consider replacing “for Sysmex hematology analyzer” with “for the Sysmex hematology analyzer ..”

• 269: estimate sensitivity =estimated sensitivity

• 272-273: obtaining a positive result in a diseased person than in a non-diseased one

• 274: a positive individual and a negative one

• 276:” diagnosis. Because” = ” diagnosis because”

• 282-283: “diagnostic accuracy ….” This sentence is not clear , consider replacing it with “ the microscopic diagnosis of malaria tends to be less sensitive in low parasitaemia and has been reported as 70% in pregnancy-associated malaria [36].

• 287-288: “which stated a sensitivity…” consider “ which reported the sensitivity of PCR to be 98%”

• 291: ….the gold standard was shown to have a higher ….

Reviewer #3: The manuscript followed sound methodological and statistical strategies, but has some typological and grammatical errors that need to be corrected to make it clear and easy to read. The discussion did not include the "and so what" of their findings and the implications of the findings.

6. PLOS authors have the option to publish the peer review history of their article (what does this mean?). If published, this will include your full peer review and any attached files.

Reviewer #1: **Yes: **Adugna Abera

Reviewer #2: **Yes: **AHMED A. ADEEL

Reviewer #3: **Yes: **Dr. Alberta Amu

---

## [Author Response · Author response to Decision Letter 0]

23 Nov 2023

We tried to corrected the comments and incorporated in the manuscript

---

## [Decision Letter · Decision Letter 1]

18 Dec 2023

Diagnostic role of Sysmex hematology analyzer in the detection of malaria: A systematic review and meta-analysis

PONE-D-23-30250R1

Dear Dr. Zewudu Mulatie ,

We’re pleased to inform you that your manuscript has been judged scientifically suitable for publication and will be formally accepted for publication once it meets all outstanding technical requirements.

Kind regards,

Bosco Bekiita Agaba, PhD

Academic Editor

PLOS ONE

Additional Editor Comments (optional):

Reviewers' comments:

Reviewer's Responses to Questions

**Comments to the Author**

1. If the authors have adequately addressed your comments raised in a previous round of review and you feel that this manuscript is now acceptable for publication, you may indicate that here to bypass the “Comments to the Author” section, enter your conflict of interest statement in the “Confidential to Editor” section, and submit your "Accept" recommendation.

Reviewer #1: All comments have been addressed

Reviewer #3: All comments have been addressed

2. Is the manuscript technically sound, and do the data support the conclusions?

Reviewer #1: Yes

Reviewer #3: (No Response)

3. Has the statistical analysis been performed appropriately and rigorously? 

Reviewer #1: Yes

Reviewer #3: (No Response)

4. Have the authors made all data underlying the findings in their manuscript fully available?

Reviewer #1: Yes

Reviewer #3: (No Response)

5. Is the manuscript presented in an intelligible fashion and written in standard English?

Reviewer #1: Yes

Reviewer #3: (No Response)

6. Review Comments to the Author

Reviewer #1: The authors have responded positively to all the comments provided. Now, from my opinion the manuscript is ready for publication.

Reviewer #3: (No Response)

7. PLOS authors have the option to publish the peer review history of their article (what does this mean?). If published, this will include your full peer review and any attached files.

Reviewer #1: **Yes: **Adugna Abera, Malaria and Neglected Tropical Diseases Research Team, Ethiopian Public Health Institute, Addis Ababa, Ethiopia

Reviewer #3: **Yes: **Dr. Alberta Amu

---

## [Editor Report · Acceptance letter]

20 Jun 2024

PONE-D-23-30250R1 

PLOS ONE

Dear Dr. Mulatie, 

I'm pleased to inform you that your manuscript has been deemed suitable for publication in PLOS ONE. Congratulations! Your manuscript is now being handed over to our production team.

Kind regards, 

on behalf of

Dr. Bosco Bekiita Agaba 

Academic Editor

PLOS ONE